# Thyroid Collision Tumors: The Presence of the Medullary Thyroid Carcinoma Component Negatively Influences the Prognosis

**DOI:** 10.3390/diagnostics13020285

**Published:** 2023-01-12

**Authors:** Ion Negura, Victor Ianole, Mihai Danciu, Cristina Preda, Diana Gabriela Iosep, Radu Dănilă, Alexandru Grigorovici, Delia Gabriela Ciobanu Apostol

**Affiliations:** 1Pathology Department, Grigore T. Popa University of Medicine and Pharmacy Iasi, 700115 Iasi, Romania; 2Sf. Spiridon Emergency Clinical Hospital Iasi, 700111 Iasi, Romania; 3Endocrinology Department, Grigore T. Popa University of Medicine and Pharmacy Iasi, 700115 Iasi, Romania; 4Faculty of Medicine, Grigore T. Popa University of Medicine and Pharmacy Iasi, 700115 Iasi, Romania; 5Surgical Department, Sf. Spiridon Emergency Clinical Hospital Iasi, 700111 Iasi, Romania

**Keywords:** collision tumor, simultaneous tumor, thyroid, papillary thyroid carcinoma, medullary thyroid carcinoma, grading system

## Abstract

Thyroid collision tumors (TCTs) are rare pathological findings, representing <1% of thyroid cancers. This study aimed to compare the main pathological features of TCTs containing medullary thyroid carcinoma (MTC) and papillary thyroid carcinoma (PTC) components with MTC-only tumors and PTC-only tumors. Methods: The retrospective study included 69 cases diagnosed with TCTs (with simultaneous MTC and PTC components), MTC and PTC. All tumors were comparatively assessed for the classical histopathological prognostic features, including a new grading system for MTC. Results: The main component of TCTs had more frequent microscopic extrathyroidal extension (mETE) (*p* = 0.000), lymphovascular invasion (LVI) (*p* = 0.000), perineural invasion (PNI) (*p* = 0.044), and lymph node metastasis (*p* = 0.042). Additionally, the TCTs’ MTC component presented with more frequent LVI (*p* = 0.010). Comparing TCTs’ MTC and PTC components with MTC-only tumors and PTC-only tumors revealed that only the TCTs’ MTC components had statistically significant more frequent mETE (*p* = 0.010) than MTC-only tumors. When applied to the MTC component of TCTs, the pathological parameters of the new grading system of MTC showed no correlations with other microscopic or clinical aspects. Conclusion: Using classical pathological prognostic features, the comparative analysis revealed that the main TCTs’ component was more aggressive than the minor one. Contrary to PTCs, in TCTs, the medullary component was more aggressive than the papillary one, but also more aggressive than MTC-only tumors.

## 1. Introduction

Thyroid collision tumors (TCTs) are rare entities, described as the simultaneous presence of more than one morphologically distinct tumor developing within the same organ, separated by non-neoplastic tissue, without a histological mixture of the different tumor cells [1,2,3]. Collision tumors can involve various regions and organs throughout the body, with cases reported in the kidney, ovary, lung, colon, and stomach [4,5,6,7,8], but they are a rare entity in the thyroid gland, representing less than 1% of all thyroid malignancies [1,9,10]. The most frequent association encountered in TCTs is the synchronous presence of medullary thyroid carcinoma (MTC) and papillary thyroid carcinoma (PTC) components [1,11]. Still, other combinations have been reported in the literature, including squamous cell carcinoma (SCC) and PTC, PTC and follicular thyroid carcinoma (FTC), Hurthle cell thyroid carcinoma (HTC) associated with PTC and poorly differentiated thyroid carcinoma (PDTC), and HTC associated with MTC [12,13,14,15].

Collision tumors with MTC and PTC components are encountered more frequently in female patients, with an F/M ratio of two to one, with the majority of cases presenting in the fifth to seventh decades of life [1,16]. Very often, after clinical and paraclinical investigations, only one component of these collision tumors is recognized, and the other component is an incidental histopathological finding [17]. It is important to mention each component in the histopathological report and to include all high-risk features: microscopic extrathyroidal extension (mETE), multifocality, perineural invasion (PNI), lymphovascular invasion (LVI), and lymph node metastasis (LNM) [18], because a complete report covering all these microscopic aspects for each component of the tumor is critical for therapy [19]. Recently, a grading system for MTC which includes mitotic activity/2 mm^2^, Ki67 proliferation index and tumor necrosis was proposed [20]. According to this grading system, MTC is considered high-grade if presents at least one of the following microscopic aspects: tumor necrosis, ≥5 mitoses/2 mm^2^ or Ki67 proliferation index ≥5% [20].

In the literature, as a consequence of the scarcity of these thyroid tumors, there is uncertainty regarding what is the essential feature of TCTs for therapy (the component with more aggressive behavior or the component having a higher stage at diagnosis) [1,11,12]. Therefore, we aimed to better understand the pathological differences between collision tumors and tumors of the thyroid with only one component. We retrospectively analyzed TCTs with MTC and PTC components and compared these components between them and with a similar number of PTC-only and MTC-only thyroid carcinomas with the same pathological T category (pT).

## 2. Materials and Methods

### 2.1. Study Group

The study group included 35 cases of TCTs, from which 23 cases with MTC and PTC components were subtracted in order to be compared with 23 cases of MTC-only tumors and 23 cases of PTCs. The inclusion criterion for PTC-only and MTC-only thyroid carcinomas was an equal pathological T category compared to the TCTs’ components. The database included the following information about the patients: age at diagnosis (younger than 55 years and older than 55 years), gender, location of the tumor (right thyroid lobe, left thyroid lobe, isthmus), and risk stratification factors, such as: pre-operative serum calcitonin levels (4 groups: 11–99 pg/mL, 100–499 pg/mL, 500–999 pg/mL, ≥1000 pg/mL), tumor size (<10 mm, 10–40 mm, and >40 mm), growth patterns, amyloid presence, focality of the tumor (≥2 foci), mitotic activity/2 mm^2^, Ki67 proliferation index (%) and tumor necrosis (present/absent) for TCTs’ MTC component and MTC-only tumors, mETE (presence of malignant cells in peri-thyroidal soft tissues including skeletal muscle, adipose tissue, vessels, and nerve bundles), LVI, PNI, LNM, coexisting thyroid pathology, pathologic T category (pT) and American Joint Committee of Cancer (AJCC) stage (8th edition, 2017) [21].

The research was performed following the ethical standards of the Helsinki declaration regarding the patients’ informed consent for the use of medical information for scientific purposes. The Ethics Committee of the “Sf. Spiridon” Clinical Emergency Hospital County Iasi approved the research project.

### 2.2. Pathological Evaluation

According to our internal protocol, the post-operatory thyroid specimen was fixed in 10% neutral buffered formalin and grossed by cutting transverse sections at 5 mm intervals. If a tumor was less than 3 cm, the tumor was entirely processed; if the tumor was larger than 3 cm, a minimum of 2 blocks per centimeter were mandatory. The capsule (if present) was completely sampled, regardless of tumor diameter. Tumor-free thyroid lobe and isthmus were sampled with one block/cm of thyroid tissue. All resected lymph nodes were submitted for microscopic evaluation.

In this retrospective study, the hematoxylin and eosin (HE), trichromic stains (Congo red, van Gieson) and immunohistochemistry (IHC) slides for all cases were independently re-evaluated by two senior pathologists according to the latest American Joint Committee of Cancer (8th edition, 2017) [21] and WHO Classification of Tumours of Endocrine Organs (4th Edition, 2017) [22], and restaged, where necessary. For immunophenotype confirmation, we used the following immunohistochemical markers: calcitonin, chromogranin, synaptophysin, CD56, thyroglobulin, HBME1, Galectin-3 and CK19.

### 2.3. Statistical Analysis

The statistical analysis was performed using Microsoft Office Excel and SPSS V.26-SPSS Inc., IBM Corporation, Chicago, IL, USA. The Chi-squared test and Fisher’s exact test were applied to assess the correlations between five elements considered to be essential for risk stratification (lymphovascular invasion, multifocality, microscopic extrathyroidal extension, perineural invasion, and lymph node metastasis), as well as patients’ clinicopathological aspects (age at diagnosis, gender, pre-operative serum calcitonin levels, mitotic activity/2 mm^2^, Ki67 proliferation index and tumor necrosis for TCTs’ MTC component and MTC-only tumors, tumor size, thyroid location, growth patterns, stromal amyloid, pathologic T category, AJCC stage, and coexisting thyroid pathology). For the comparative analysis, the independent t-test was used to compare the variables (elements considered essential for risk stratification and clinicopathological aspects of the patients) of the two components of the TCTs between them and with similar PTC-only and MTC-only thyroid tumors. Every test was 2-sided, and a *p*-value < 0.05 (5%) was considered statistically significant.

## 3. Results

### 3.1. Epidemiological and Pathological Characteristics of Collision Tumors Cohort

We retrospectively examined the epidemiological and pathological characteristics of 35 cases of TCT (10 males, 25 females, M/F = 0.40), with a mean age at diagnosis of 61.54 years (range 40–77 years), diagnosed in our hospital between 2008 and 2022 (Table 1).

When comparing the main component with the minor one (Table 2), we observed a statistically significant difference in the following pathological characteristics: diameter, microscopic extrathyroidal extension (mETE), LVI, PNI, LNM, pathologic T category and AJCC stage, pointing out that the main component was more aggressive.

### 3.2. Pathological Characteristics of TCTs with MTC and PTC Components and Comparison with MTC-Only Tumors and PTCs

Out of 35 TCT cases diagnosed over a 15 year period, 23 cases (65.71%) were diagnosed as collision tumors with MTC and PTC components (7 males, 16 females, M/F = 0.43; mean age of 61.6 years, range 42–77 years) (Table 1).

#### 3.2.1. Comparison between MTC and PTC Components of TCTs

When comparing the MTC and PTC components (Figure 1) (Table 3), the following pathological characteristics, diameter, LVI, pathologic T category and AJCC stage, had statistically significant differences, pointing out that the MTC component was more aggressive.

#### 3.2.2. Analysis of Prognostic Factors of MTC Component of TCTs

To evaluate the correlations between pathological features with prognostic value, we assessed the associations between age at diagnosis, gender, pre-operative serum calcitonin levels, tumor size, location, growth pattern, stromal amyloid, mitotic activity/2 mm^2^, Ki67 proliferation index, tumor necrosis pathologic T category, AJCC stage, coexisting thyroid pathology, and five elements considered to be essential for risk stratification, namely, PNI, LVI, mETE, multifocality, and LNM (Appendix A).

Statistically significant correlations were observed between LVI (present vs. absent) and the pre-operative serum calcitonin levels (*p* = 0.005), with a strong correlation for calcitonin levels ≥1000 pg/mL. Additionally, LVI had statistically significant correlations with pathologic T category (*p* = 0.001), AJCC stage (*p* = 0.000), tumor size (*p* = 0.002), and presence of amyloid (*p* = 0.036).

Lymph node metastases due to TCTs’ MTC component (present vs. absent) had statistically significant correlations with the pre-operative serum calcitonin levels (*p* = 0.005), with a strong correlation at calcitonin levels ≥1000 pg/mL. Additionally, LNM had statistically significant correlations with pathologic T category (*p* = 0.005), AJCC stage (*p* = 0.000), and tumor size (*p* = 0.006).

Microscopic extrathyroidal extension due to TCTs’ MTC component (present vs. absent) had statistically significant correlations with the pathologic T category (*p* = 0.025) and AJCC stage (*p* = 0.044).

Focality of the TCTs’ MTC component (unifocal vs. multifocal) had statistically significant correlations with the gender of the patients (*p* = 0.020).

#### 3.2.3. Analysis of Prognostic Factors of MTC-Only Tumors 

LVI (present vs. absent) had statistically significant correlations with tumor size (*p* = 0.032) and with AJCC stage (*p* = 0.023). Statistically significant correlations between the focality of the tumors (unifocal vs. multifocal) and growth patterns were present (*p* = 0.024) (Table 4).

#### 3.2.4. TCTs’ MTC Component vs. MTC-Only Tumors

The main clinical and pathological features of TCT’s MTC component are presented in Table 1. The growth patterns of the TCTs’ MTC component were variable: they were solid in 15 cases (65.21%) (Figure 2A), insular in 4 cases (17.39%), trabecular in 3 cases (13.04%), and lobular in 1 case (4.34%). By contrast, MTC-only tumors had only three growth patterns: they were solid in 19 cases (82.60%), and insular and trabecular in 2 cases (8.69%) each. Amyloid was identified in 13 cases (56.52%) (Figure 2B) of TCTs with MTC components, compared to 12 (52.17%) cases of MTC-only tumors.

TCTs’ MTC component and MTC-only tumors were multifocal in three cases (13.04%), each. Microscopic ETE due to TCTs’ MTC component was present in eight cases (34.78%), with LVI in eleven cases (47.82%) (Figure 2C), and PNI in three cases (13.04%), while MTC-only cases presented with mETE and PNI in one case (4.34%) each, and with LVI in eight cases (34.78%).

Based on mitotic activity/2 mm^2^, Ki67 proliferation index and tumor necrosis, only one (4.34%) TCT’s MTC component was high-grade compared to four cases (17.39%) of MTC-only tumors.

Lymph node metastasis due to TCTs’ MTC component was present in seven cases (30.43%) (Figure 2D). In comparison, there were six cases (26.08%) with lymph node metastasis due to MTC-only tumors. IHC tests were performed to demonstrate the MTC component, including calcitonin (Figure 2E), chromogranin (Figure 2F), and synaptophysin, when necessary.

When comparing the TCTs’ MTC component with MTC-only tumors, (Table 5), only mETE due to TCTs’ MTC components had a statistically significant difference, pointing out that MTC component was more aggressive. 

#### 3.2.5. Analysis of Prognostic Factors of PTC Component of TCTs

Statistically significant correlations (Appendix A) were observed between LVI due to TCTs’ PTC component and pathologic T category (*p* = 0.034). Additionally, LVI had statistically significant correlations with tumor size (*p* = 0.034). mETE, due to TCTs’ PTC component, had statistically significant correlations with tumor size (*p* = 0.034) and pathologic T category (*p* = 0.034). TCTs’ PTC component focality had statistically significant correlations with the growth patterns (*p* = 0.014).

#### 3.2.6. Analysis of Prognostic Factors of PTC-Only Tumors

Only LVI had statistically significant correlations (Appendix A) with age at diagnosis (*p* = 0.021), tumor size (*p* = 0.002), growth patterns (*p* = 0.048) and pathologic T category (*p* = 0.011). 

#### 3.2.7. TCTs’ PTC Component vs. PTC-Only tumors

The main clinical and pathological features of TCT’s PTC component are presented in Table 1.

TCTs’ PTC component and PTC-only tumors had only two major growth patterns, with 12 cases (52.17%) of a follicular growth pattern (Figure 3A) and 11 cases (47.82%) of a conventional growth pattern (Figure 3B) for TCTs’ PTC component. PTC-only tumors had conventional growth patterns in 14 cases (60.86%) and follicular growth patterns in 9 cases (39.13%).

IHC tests to demonstrate the PTC component, including CK19 (Figure 3C), Galectin-3 (Figure 3D), HBME1 (Figure 3E), and CD56 (Figure 3F), were performed when necessary. No statistically significant differences (Appendix A) between TCTs’ PTC component and PTC-only tumors were observed.

## 4. Discussion

TCT represents a particular category of thyroid tumors with different pathogenesis compared to mixed or composite thyroid tumors [1,23]. These terms were used interchangeably in the thyroid gland to designate multiple concomitant tumors composed of parafollicular C cells and follicular cellular elements [24]. However, these terms are not similar and indicate distinct neoplastic entities [25]. Collision tumors are synchronous, morphologically distinct tumors, developing within the same organ and separated by non-neoplastic tissue, without a histological mixture of the different tumor cells [1,13]. Mixed tumors, which, according to WHO 2017 [22] are synonymous to composite tumors, are malignant thyroid tumors with histopathological and immunohistochemical aspects of follicular and parafollicular C cells that coexist intermixed in the same thyroid tumor [26]. The source of mixed/composite tumors is a common driver mutation that induces divergent cellular differentiation, which gives rise to two or more different intermingled cellular populations [27].

In mixed medullary-follicular thyroid carcinoma (MMFTC), because of a common cell of origin, tumor cells are positive (individual cells or separate cell clusters) for both thyroglobulin and calcitonin within the same lesion [28], while TCTs composed of MTC and follicular components are positive for calcitonin only in the MTC component and thyroglobulin only in the follicular component, as a consequence of different pathogeneses (origin from two or more different cell progenitors separated by non-neoplastic tissue) [16]. Multiple hypotheses tempted to explain the development of TCTs. The first is the “chance theory”, which suggests a separate origin for components of TCTs and the synchronous development of the components by chance [13]. A second hypothesis assumes that the initial tumor can modify the microenvironment to promote the occurrence and development of another tumor [3]. Another theory suspects the development of two or more separate driver mutations in the same stem cell that induce the occurrence of two or more independent tumors [13]. Nonetheless, there is still uncertainty regarding the pathogenesis of TCTs, as many case reports of TCT with molecular analysis performed to identify the cell of origin for these tumors communicate different and conflicting results [1,11,16,29,30].

Different studies published results indicate that some pathological features, namely the focality of the tumor, extrathyroidal extension, lymphovascular invasion, perineural invasion, and lymph node metastasis, can be crucial prognostic factors in thyroid carcinomas [31,32,33,34,35]. In addition to these high-risk pathological features, in the case of MTC, ≥5 mitoses/2 mm^2^, Ki67 proliferation index ≥ 5% or tumor necrosis are considered poor prognostic factors, and when at least one is microscopically identified, the tumor is considered high-grade [20]. To our knowledge, this is the first study which analyzes the MTC component of TCTs using the recently proposed International Medullary Thyroid Carcinoma Grading System [20]. Additionally, considering the rarity of this entity (1% of the entire thyroid neoplastic pathology) [10], we consider that the selected TCT cases are relevant. With 23 cases of TCT with MTC and PTC components, our series is the third-largest one described in literature, after those published by Appetecchia et al. in 2019 (183 patients) [1] and Biscolla et al. in 2004 (27 patients) [29].

The present study mainly aimed to analyze high-risk histopathological features (mETE, multifocality, LVI, and LNM) of each TCT’s MTC and PTC components and to compare them with those of similar MTC-only and PTC-only thyroid tumors in order to document the differences between these separate entities. We also compared the high-risk histopathological features of each component to point out which one is more aggressive. Eventually, a univariate analysis was performed to document the correlations (for each TCT component with MTC-only and PTC-only tumors) between the high-risk histopathological features and the patients’ main pathological features.

When comparing the high-risk histopathological features of TCTs’ components with similar thyroid carcinomas, only TCTs’ MTC component had a statistically significant more aggressive behavior than MTC-only tumors. In opposition to our study, Biscolla et al. found no different behavior of TCTs’ MTC component compared to MTC-only tumors when analyzing the epidemiologic, clinical, and pathologic features, stating that the association of PTC component in a TCT does not influence the outcome of the MTC [29].

Regarding the behavior of the components present in the 23 cases of TCT with MTC and PTC components, when comparing the high-risk histopathological features of the two components between them, the MTC components were more aggressive, presenting with more frequent LVI. Additional statistically significant differences, implying a more aggressive behavior of the MTC components, were present, when comparing the diameter and AJCC stage, pointing out that the TCTs’ main components were more aggressive in our study.

MTC was invariably the main component in our case series (20/23 cases) (86.95%). The results regarding the behavior of the two components (MTC and PTC) reported in our study are similar with the results of Appetecchia et al. and Thomas et al. [1,11]. Both our study and Thomas et al.’s study found a more aggressive behavior of the MTC component than the PTC one, observations sustained by the larger mean tumor size, higher frequency of ETE, LVI, more frequent LNM due to the MTC component, and a higher stage compared to PTC component [11].

The female preponderance (18F/9M, F/M = 2) that we found in our study was similar to the results of Biscolla et al. (16F/7M F/M = 2.28) [29] and Appetecchia et al. (105F/78M F/M = 1.34) [1], while the case series published by Thomas et al. had a slight male preponderance (10F/11M F/M = 0.90) [11]. In contrast to our study, where the mean age of the patients was 61.7 years (range 42–77 years), the studies published by Appetecchia et al., Biscolla et al. and Thomas et al. communicate a lower mean age of 56.2 (range 16–84 years) [1], 49.9 (range 27–76 years) [29], and 45.3 years (range 26–77 years) [11], respectively. On the other hand, the study published by Biscolla et al. had only six cases (22.22%) with both components involving the same thyroid lobe [29], while in our study there were twelve cases (52.17%), similar to Thomas et al. (11/21–52.38%) [11]. Micro-PTC was present in 20/23 cases (86.95%) of TCT with MTC and PTC components in our study, a percentage which is close to those reported by Appetecchia et al., with 148/183 (81%) [1], Biscolla et al., with 21/27 cases (77.77%) [29], and Thomas et al., with 18/21 cases (85.71%) [11]. In most of our patients, both components were located in the same thyroid lobe (52.17%), similar to Thomas et al. (52.38%) [11]. This supports the theory suggesting that the initial tumor can modify the microenvironment to promote the occurrence and development of another tumor. In contrast, the study performed by Biscolla et al. had 21 of 27 cases (77%) with MTC and the PTC located in different thyroid lobes [29], suggesting that these two carcinomas are two different diseases with no interconnection.

The disparate results reported by our study when compared to those published by Biscolla et al. and Thomas et al. may be explained by different grossing protocols, particularly of the uninvolved thyroid lobe by a tumor. Another reason might be the existence of hard-to-recognize MTC variants (oncocytic, follicular, and pseudopapillary subtypes), which may be incorrectly diagnosed as HTC, FTC, or PTC [36].

There was no statistically significant difference, in our study, between TCTs’ MTC components and MTC-only cases when the high-risk pathological features (≥5 mitoses/2 mm^2^, Ki67 proliferation index ≥ 5% or tumor necrosis) were compared. Contrary to the study published by Xu et al. [20], we observed no correlations between these high-risk pathological features and PNI, LVI, mETE, multifocality, and LNM.

Our study had some limitations. For instance, we could not correlate the pathological data with the cytological examination for all our patients. Yet, the management of these cases was not negatively influenced. Additionally, no molecular tests were performed.

Because TCT is rare, treatment protocols are poorly defined; some authors assume that the two components of TCT should be treated as two distinct concomitant primaries [3], while others argue that the treatment should focus on the component with the most aggressive features [1,12]. The treatment protocol of TCTs is complex and needs to consider the stage of the tumor but also clinical and morpho-pathological signs of aggressiveness for each component; thus, these tumors are best managed by an interdisciplinary team that can elaborate an individualized therapy scheme for these patients [37]. In the case of TCTs with MTC and PTC components, total thyroidectomy is the recommended surgical procedure because of the aggressive behavior of the MTC component, which can be multicentric and involve both lobes at the same time [38]. Additionally, the increased incidence of tumor LVI in TCTs with an MTC component requires neck dissection with lymphadenectomy in the central and lateral compartment for staging and locoregional tumor control [1,29,39]. T category of the main component and pre-operative serum calcitonin levels are the significant factors guiding prophylactic neck lymph node dissection in clinical and paraclinical node-negative cases [40]. Adjuvant radioactive iodine therapy is required if the TCTs PTC component is grouped in a high or intermediate-risk category in the risk stratification scheme [41,42]. In the study published by Biscolla et al., the outcome of the patients with TCT having an MTC component was not affected by the PTC component or radioiodine treatments [29]. One possible explanation might be that the TCTs PTC component was an incidental m-PTC without high-risk features in most of his cases. In our study, three cases of the PTC component with a maximum diameter over 20 mm but not more than 40 mm (T2 category) showed variable proportions of high-risk features, such as multifocality, microscopic ETE, LVI, and lymph node metastasis. These different therapeutic strategies that consider many clinical and morpho-pathological aspects impose the need for a complete microscopic evaluation for each TCTs component, including high-risk aspects, to manage this particular category of patients better.

## 5. Conclusions

MTC and PTC are the most common components of TCTs. The prognosis is influenced by the component with the higher stage and the most aggressive behavior, which most frequently is the MTC component, as in our study and in most cases reported in literature. Regarding the high-risk histopathological features, only mETE due to TCTs’ MTC component proved to be statistically different compared to MTC-only tumors, pointing out that TCTs’ MTC component may be more aggressive than MTC-only tumors. When applied to MTC component of TCTs, the pathological parameters of the recently proposed International Medullary Thyroid Carcinoma Grading System showed no correlations with other microscopic or clinical aspects.

Although our study is one of the largest published to date, further studies are necessary to analyze the differences and the outcome of patients with TCTs composed of MTC and PTC compared to MTC-only and PTC-only cases, to better understand the pathogenesis, behavior, and optimal management of these cases.

## Figures and Tables

**Figure 1 diagnostics-13-00285-f001:**
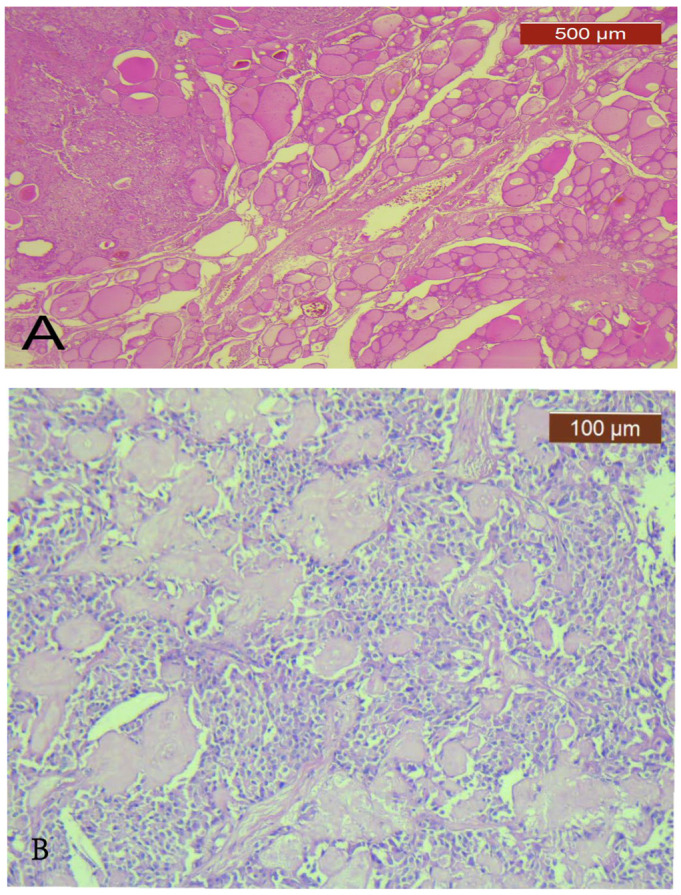
Collision tumor. (**A**) low power view to show a TCT comprising MTC (**upper left**) and micro-PTC (**lower right**) components in the same thyroid lobe separated by non-tumoral thyroid tissue (HE, ×25); (**B**) higher magnification of the MTC component showing a classical MTC with amyloid deposits (HE, ×100); (**C**) higher magnification of a conventional PTC component (HE, ×200).

**Figure 2 diagnostics-13-00285-f002:**
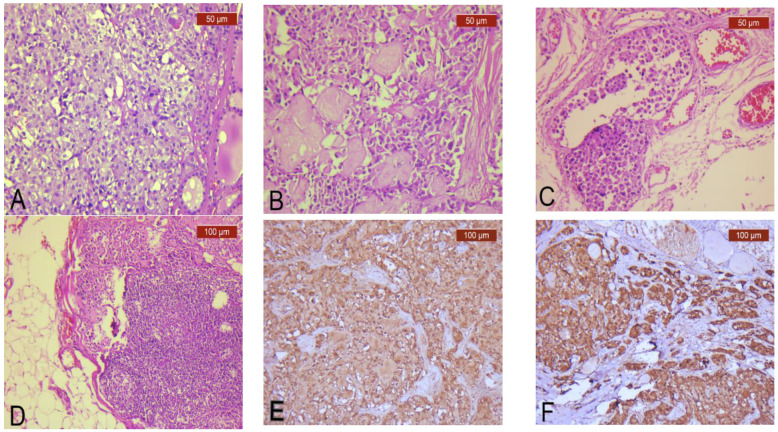
Pathological features of TCTs’ MTC components. (**A**) classical solid MTC growth pattern (HE, ×200); (**B**) stromal amyloid present in the MTC component (HE, ×200); (**C**) lymphovascular invasion due to the MTC component (HE, ×200); (**D**) lymph node metastasis due to the MTC component (HE, ×100); strong and diffuse positivity of the MTC component for (**E**) calcitonin (IHC, anti-calcitonin antibody, ×100) and (**F**) chromogranin (IHC, anti-chromogranin antibody, ×100).

**Figure 3 diagnostics-13-00285-f003:**
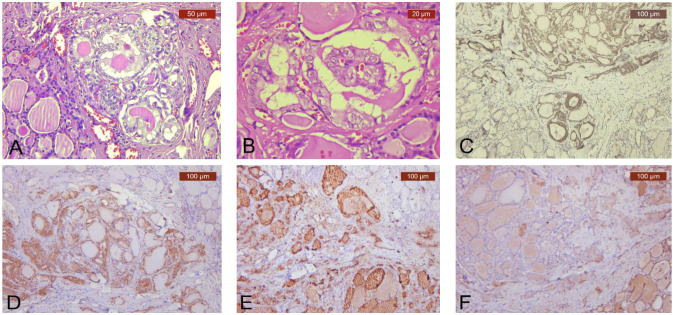
Histo-morphological features of TCTs’ PTC components. (**A**) Follicular PTC growth pattern (HE, ×200); (**B**) conventional PTC growth pattern (HE, ×400); strong and diffuse positivity of PTC component for (**C**). CK19 (IHC, anti-CK19 antibody, ×100), (**D**) galectin-3 (IHC, anti-galectin-3 antibody, ×100), (**E**) HBME1 (IHC, anti-HBME1 antibody, ×100); (**F**) total negativity for CD56 (IHC, anti-CD56 antibody, ×100).

**Table 1 diagnostics-13-00285-t001:** Clinical and histopathological aspects of the TCT cases.

Parameter		Number of Cases
Gender	M	10 (28.57%)
	F	25 (71.42%)
Age at diagnosis	<55	7 (20%)
	≥55	28 (80%)
Location of the TCTs components	RTL	29 (82.85%)
	LTL	23 (65.71%)
	Isthmus	3 (8.57%)
Associated components	MTC + PTC	23 (65.71%)
	FTC + PTC	5 (14.28%)
	HTC + PTC	4 (11.42%)
	ATC + PTC	2 (5.71%)
	PDTC + PTC	1 (2.85%)
mETE	Main component	15 (42.85%)
	Minor component	2 (5.71%)
LVI	Main component	23 (65.71%)
	Minor component	2 (5.71%)
PNI	Main component	4 (11.42%)
	Minor component	0 (0%)
Multifocality	Main component	5 (14.28%)
	Minor component	9 (25.71%)
LNM	Main component	9 (25.71%)
	Minor component	3 (8.57%)
Pathological N category		
pNx		14 (40%)
pN0		10 (28.57%)
pN1a	Main component	3 (8.57%)
	Minor component	2 (5.71%)
pN1b	Main component	6 (17.14%)
	Minor component	1 (2.85%)
Pathological T category		
pT1	Main component	12 (34.28%)
	Minor component	35 (100%)
pT2	Main component	9 (25.71%)
	Minor component	0 (0%)
pT3	Main component	11 (31.42%)
	Minor component	0 (0%)
pT4	Main component	3 (8.57%)
	Minor component	0 (0%)
AJCC stage		
I	Main component	19 (54.28%)
	Minor component	33 (94.28%)
II	Main component	7 (20%)
	Minor component	2 (5.71%)
III	Main component	2 (5.71%)
	Minor component	0 (0%)
IV	Main component	7 (20%)
	Minor component	0 (0%)
Coexistent thyroid pathologies		
Colloid goiter		18 (51.42%)
Nodular goiter		11 (31.42%)
Hashimoto thyroiditis		6 (17.14%)

Abbreviations: M male, F female, LTL left thyroid lobe, RTL right thyroid lobe, mETE microscopic extrathyroidal extension, LVI lymphovascular invasion, PNI perineural invasion, LNM lymph node metastasis, PTC papillary thyroid carcinoma, MTC medullary thyroid carcinoma, FTC follicular thyroid carcinoma, HTC Hürthle cell thyroid carcinoma, PDTC poorly differentiated thyroid carcinoma, ATC anaplastic thyroid carcinoma

**Table 2 diagnostics-13-00285-t002:** Pathological characteristics of the TCTs’ main components compared to their minor components (Chi-squared test).

		Mean Value	*p*-Value
Diameter (mm)	M	35.486	0.000
	m	3.946	
Focality of the tumor (Unifocal/Multifocal)	M	1.860	0.238
	m	1.740	
mETE (Present/Absent)	M	1.430	0.000
	m	1.060	
LVI (Present/Absent)	M	1.660	0.000
	m	1.060	
PNI (Present/Absent)	M	1.110	0.044
	m	1.000	
Lymph node metastasis (Present/Absent)	M	1.430	0.042
	m	1.140	
Pathologic T category (T1/T2/T3/T4)	M	2.140	0.000
	m	1.000	
AJCC stage (I/II/III/IV)	M	1.910	0.000
	m	1.060	

*p*-value < 0.05 was considered to be statistically significant. Abbreviations: LTL left thyroid lobe, RTL right thyroid lobe, mETE microscopic extrathyroidal extension, LVI lymphovascular invasion, PNI perineural invasion, M main component, m minor component.

**Table 3 diagnostics-13-00285-t003:** Pathological characteristics of the TCTs’ MTC components compared to PTC components (Chi-squared test).

		Mean Value	*p*-Value
Diameter (mm)	MTC	27.435	0.001
	PTC	6.174	
Thyroid site (LTL/RTL/Isthmus)	MTC	2.570	0.485
	PTC	2.430	
Focality of the tumor (Unifocal/Multifocal)	MTC	1.870	0.275
	PTC	1.740	
mETE (Present/Absent)	MTC	1.350	0.088
	PTC	1.130	
LVI (Present/Absent)	MTC	1.480	0.010
	PTC	1.130	
PNI (Present/Absent)	MTC	1.130	0.307
	PTC	1.040	
Lymph node metastasis (Present/Absent)	MTC	1.470	0.130
	PTC	1.200	
pathologic T category (T1/T2/T3/T4)	MTC	1.740	0.011
	PTC	1.130	
AJCC stage (I/II/III/IV)	MTC	1.960	0.004
	PTC	1.090	

*p*-value < 0.05 was considered to be statistically significant. Abbreviations: LTL left thyroid lobe, RTL right thyroid lobe, mETE microscopic extrathyroidal extension, LVI lymphovascular invasion, PNI perineural invasion, MTC medullary thyroid carcinoma, PTC papillary thyroid carcinoma.

**Table 4 diagnostics-13-00285-t004:** Pathological characteristics of the MTC-only tumors in relation to the histopathological prognostic features (Chi-squared test).

Pathological Characteristics	Tumor Focality		*p*-Value	LVI		*p*-Value
	U (#20)	M (#3)		P (#8)	A (#15)	
Tumor size (mm)						
<10 mm	4 (20%)	1 (33.3%)	0.124	0 (0%)	5 (33.3%)	0.032
10–40 mm	11 (55%)	0 (0%)		3 (37.5%)	8 (53.3%)	
>40 mm	5 (25%)	2 (66.7%)		5 (62.5%)	2 (13.3%)	
Growth pattern						
Insular	2 (10%)	0 (0%)	0.024	1 (12.5%)	1 (6.7%)	1.000
Solid	18 (90%)	1 (33.3%)		6 (75%)	13 (86.7%)	
Trabecular	0 (0%)	2 (66.7%)		1 (12.5%)	1 (6.7%)	
AJCC stage						
I	12 (60%)	1 (33.3%)	0.163	3 (37.5%)	10 (66.7%)	0.023
II	4 (20%)	0 (0%)		1 (12.5%)	3 (20%)	
III	2 (10%)	0 (0%)		0 (0%)	2 (13.3%)	
IV	2 (10%)	2 (66.7%)		4 (50%)	0 (0%)	

*p*-value < 0.05 was considered to be statistically significant Abbreviations: LVI, lymphovascular invasion; P, present; A, absent; U, unifocal; M, multifocal; #, number of cases.

**Table 5 diagnostics-13-00285-t005:** Pathological characteristics of the TCTs’ MTC components compared to MTC-only tumors.

		Mean Value	t-Test Value	Degrees of Freedom (df)	*p*-Value
Age at diagnosis (years)	MTCc	61.700	1.110	44.000	0.273
	MTC-only	57.960			
Sex (M/F)	MTCc	1.300	0.660	44.000	0.513
	MTC-only	1.220			
Diameter (mm)	MTCc	27.430	0.567	44.000	0.574
	MTC-only	23.700			
Thyroid site (LTL/RTL/Isthmus)	MTCc	1.570	0.000	44.000	1.000
	MTC-only	1.570			
Growth pattern (Solid/Trabecular/Insular/Lobular)	MTCc	2.740	-0.736	44.000	0.466
	MTC-only	2.910			
Amyloid (Present/Absent)	MTCc	1.570	0.290	44.000	0.773
	MTC-only	1.520			
Tumor focality (Unifocal/Multifocal)	MTCc	1.870	0.000	44.000	1.000
	MTC-only	1.870			
mETE (Present/Absent)	MTCc	1.350	2.755	29.804	0.010
	MTC-only	1.040			
LVI (Present/Absent)	MTCc	1.480	0.886	44.000	0.380
	MTC-only	1.350			
PNI (Present/Absent)	MTCc	1.130	1.036	36.221	0.307
	MTC-only	1.040			
Lymph node metastasis (Present/Absent)	MTCc	1.470	0.502	29.000	0.619
	MTC-only	1.380			
pathologic T category (T1/T2/T3/T4)	MTCc	1.740	0.152	44.000	0.880
	MTC-only	1.700			
AJCC stage (I/II/III/IV)	MTCc	1.960	0.242	44.000	0.810
	MTC-only	1.870			

*p*-value < 0.05 was considered to be statistically significant Abbreviations: M male, F female, LTL left thyroid lobe, RTL right thyroid lobe, mETE microscopic extrathyroidal extension, LVI lymphovascular invasion, PNI perineural invasion, MTCc medullary thyroid carcinoma component, MTC-only medullary thyroid carcinoma-only.

## Data Availability

Not applicable.

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
