# Peer review of "Thyroid Collision Tumors: The Presence of the Medullary Thyroid Carcinoma Component Negatively Influences the Prognosis"

_diagnostics, 2023, doi:10.3390/diagnostics13020285_

Round 1

Reviewer 1 Report

The study of Ion Negura et al presents some rare cases of thyroid collision tumors. The conclusions of the study are solid and direct, however the way the study is presented could be improved.

1.     The verses are not numbered, which makes the revision process a bit harder

2.     Section 2.1. Study group “(…) and American Joint Committee of Cancer (AJCC) stage”. – please indicate which version of AJCC staging is used.

3.     Section RESULTS:

“Only the main collision tumor components were staged in pT2, pT3, and pT4 categories, with 9 cases (25.71%) (PTC 3 cases - 8.57% and MTC, HTC, and FTC each in 2 cases - 5.71%) grouped into the pT2 category (2 multifocal PTC components), 11 cases (31.42%) (MTC 6 cases - 17.14%, FTC 3 cases - 8.57%, HTC and PDTC each in 1 case - 2.85%) in the pT3a category (1 multifocal MTC component), and 3 cases (8.57%) (ATC 2 cases - 5.71% and MTC 1 case - 2.85%) in the pT4a category (none multifocal).”

Perhaps it would be better to present this part in a table or graphical form?

What I mean is that the long descriptions of the results make the study difficult to follow. Perhaps the descriptions could be shortened?

4.     The results section:

“The difference between the diameter means of the two components was statistically significant”

Here the authors describe a value determined as “diameter means”, but later in the text this value is referred to as “means” (as I understand it), which caused some confusion for me and made the text difficult to follow. It would be beneficial for the study to screen the manuscript for this kind of mistakes

5.     Section 3.2.4. TCTs’ MTC component vs. MTC-only tumors – this section mostly presents the same results that are presented in table 4, please avoid such long descriptions, especially if there is only one characteristic that is statistically significant. This mistake is repeated throughout the whole manuscript and it makes the study difficult to follow.

6.     The discussion section:

“When comparing the high-risk histopathological features of all 35 TCTs’ components with each other, the main components were more aggressive, with more frequent mETE - t (48.269) = 3.962, p=0.000 (<0.05), LVI - t (49.381) = 6.621, p=0.000 (<0.05), PNI - t (34.000) = 2.095 p=0.044 (<0.05) and LNM - t (36.000) = 2.108, p=0.042 (<0.05). Also, statistically significant differences implying a more aggressive behavior of the main components were present, when comparing the diameter - t (35.411) = 7.456, p=0.000 (<0.05), pT category - t (34.000) = 6.733, p=0.000 (<0.05), and AJCC stage - t (36.627) = 4.156, p=0.000 (<0.05) of the two components.”

If possible, please avoid presenting the all the results in the discussion section. Present only the significant values or conclusion of the particular results

Author Response

Greta-Laura Balan, Assigned Editor - Diagnostics

Dear Greta-Laura Balan,

Thank you very much for the opportunity to respond to reviewers’ comments and revise our manuscript. We would like to address our thanks to reviewers for their constructive recommendations. Their comments were discussed with authors and the manuscript was adjusted where appropriate and provided detailed explanations for issues that needed clarification and hope the manuscript is now acceptable for publication.

Mihai Danciu

On behalf of authors

Reviewer's comments

Reviewer #1: The study of Ion Negura et al presents some rare cases of thyroid collision tumors. The conclusions of the study are solid and direct, however the way the study is presented could be improved.

1. The verses are not numbered, which makes the revision process a bit harder

Thank you for this useful observation. We added line numbering; we hope they are visible.

2. Section 2.1. Study group “(…) and American Joint Committee of Cancer (AJCC) stage”. – please indicate which version of AJCC staging is used.

Thank you for this suggestion. We added the Edition and the year of publication with references.

3. Section RESULTS:

“Only the main collision tumor components were staged in pT2, pT3, and pT4 categories, with 9 cases (25.71%) (PTC 3 cases - 8.57% and MTC, HTC, and FTC each in 2 cases - 5.71%) grouped into the pT2 category (2 multifocal PTC components), 11 cases (31.42%) (MTC 6 cases - 17.14%, FTC 3 cases - 8.57%, HTC and PDTC each in 1 case - 2.85%) in the pT3a category (1 multifocal MTC component), and 3 cases (8.57%) (ATC 2 cases - 5.71% and MTC 1 case - 2.85%) in the pT4a category (none multifocal).”

Perhaps it would be better to present this part in a table or graphical form?

What I mean is that the long descriptions of the results make the study difficult to follow. Perhaps the descriptions could be shortened?

Thank you for this useful recommendation. We removed the data from main text and presented it in Table 1.

4. The results section:

“The difference between the diameter means of the two components was statistically significant”

Here the authors describe a value determined as “diameter means”, but later in the text this value is referred to as “means” (as I understand it), which caused some confusion for me and made the text difficult to follow. It would be beneficial for the study to screen the manuscript for this kind of mistakes

Thank you for this observation. By removing the descriptive text, the confusing terms were avoided. The Tables contain the term “diameter” which is eloquent.

5. Section 3.2.4. TCTs’ MTC component vs. MTC-only tumors – this section mostly presents the same results that are presented in table 4, please avoid such long descriptions, especially if there is only one characteristic that is statistically significant. This mistake is repeated throughout the whole manuscript and it makes the study difficult to follow.

Thank you for this useful recommendation. We removed the data from main text and presented it in Table 5 (Tables were renumbered after introducing Table 1).

6. The discussion section:

“When comparing the high-risk histopathological features of all 35 TCTs’ components with each other, the main components were more aggressive, with more frequent mETE - t (48.269) = 3.962, p=0.000 (<0.05), LVI - t (49.381) = 6.621, p=0.000 (<0.05), PNI - t (34.000) = 2.095 p=0.044 (<0.05) and LNM - t (36.000) = 2.108, p=0.042 (<0.05). Also, statistically significant differences implying a more aggressive behavior of the main components were present, when comparing the diameter - t (35.411) = 7.456, p=0.000 (<0.05), pT category - t (34.000) = 6.733, p=0.000 (<0.05), and AJCC stage - t (36.627) = 4.156, p=0.000 (<0.05) of the two components.”

If possible, please avoid presenting the all the results in the discussion section. Present only the significant values or conclusion of the particular results.

Thank you for this suggestion. We consider it very important. We removed the detailed results presentation from Discussion chapter, yet keeping only the significant ones.

Reviewer 2 Report

This is a very interesting paper describing a rare subset of thyroid tumors along with their clinical manifestations. The manuscript is well written and the presented data are valuable. However, I do have some major concerns regarding statistical analysis. Should these concerns be properly addressed, I would be happy to recommend the paper for publication.

Major concern:

In Tables 1 and 2, the majority of presented data are ordinal variables (such as focality, mETE, LVI, lymph node metastases, T category, etc.). Hence, a propper way to analyze these data would be a Chi square test (or Fisher exact, where appropriate), not a t-test. A t-test is more suitable for scale measures following a normal distribution (in this case, only for tumor size and age at diagnosis). Therefore, I would suggest the authors to recalculate the p-values in Table1 and Table2 with  a correct statistical test.

Minor concern:

At the beginning of the discussion section, the authors should give a more comprehensive explanation on the differences between collision, mixed and composite tumors.

Author Response

Greta-Laura Balan, Assigned Editor - Diagnostics

Dear Greta-Laura Balan,

Thank you very much for the opportunity to respond to reviewers’ comments and revise our manuscript. We would like to address our thanks to reviewers for their constructive recommendations. Their comments were discussed with authors and the manuscript was adjusted where appropriate and provided detailed explanations for issues that needed clarification and hope the manuscript is now acceptable for publication.

Mihai Danciu

On behalf of authors

Reviewer's comments

Reviewer #2: This is a very interesting paper describing a rare subset of thyroid tumors along with their clinical manifestations. The manuscript is well written and the presented data are valuable. However, I do have some major concerns regarding statistical analysis. Should these concerns be properly addressed, I would be happy to recommend the paper for publication.

Major concern

  1. In Tables 1 and 2, the majority of presented data are ordinal variables (such as focality, mETE, LVI, lymph node metastases, T category, etc.). Hence, a propper way to analyze these data would be a Chi square test (or Fisher exact, where appropriate), not a t-test. A t-test is more suitable for scale measures following a normal distribution (in this case, only for tumor size and age at diagnosis). Therefore, I would suggest the authors to recalculate the p-values in Table1 and Table2 with a correct statistical test.

Thank you for this recommendation. We modified the above mentioned Table 1 and Table 2 (which now, after introducing a new Table 1, became Table 2 and Table 3, respectively). Now the tables present the p-value after applying Chi square test (as mentioned in the Tables captions).

Minor concern

  1. At the beginning of the discussion section, the authors should give a more comprehensive explanation on the differences between collision, mixed and composite tumors.

Thank you for this useful recommendation. We fully agree it is important for our readers. We added definitions of these entities and comprehensive explanation of the differences between them.

Round 2

Reviewer 1 Report

All my commentaries have been replied to. In my opinion the study can now be considered for publication